

# Assessment of combined serum sST2 and AFP levels in the diagnosis of hepatocellular carcinoma

Xiuxin Tang[1,*], Dong Wang[1,2,*], Tangdan Ding[1], Rongqi Lin[3], Meifang He[4], Ruizhi Wang[1,5] and Liubing Li[1]

[1] Department of Laboratory Medicine, The First Affiliated Hospital of Sun Yat-sen University, Guangzhou, Guangdong, China
[2] Department of Laboratory Medicine, Guangxi Hospital Division of The First Affiliated Hospital, Sun Yat-sen University, Nanning, Guangxi, China
[3] Department of Pharmacy, Shanghang County Hospital, Shanghang, FuJian, China
[4] Laboratory of General Surgery, The First Affiliated Hospital of Sun Yat-sen University, Guangzhou, Guangdong, China
[5] Advanced Medical Technology Center, Sun Yat-sen University, Guangzhou, Guangdong, China
[*] These authors contributed equally to this work.

Corresponding authors
Ruizhi Wang, wangrzh3@mail.sysu.edu.cn
Liubing Li, lilb8@mail.sysu.edu.cn

## ABSTRACT

**Background.** Hepatocellular carcinoma (HCC) is a common malignant tumor with high morbidity and mortality. Alpha-fetoprotein (AFP) is the most widely used diagnostic serum biomarker, but it still has limited accuracy in detecting HCC, suggesting the necessity of seeking more ideal biomarkers with high sensitivity and specificity. Soluble growth stimulation gene 2 (sST2) form of growth stimulating expression gene 2 (ST2), is expressed in various organs and can bind competitively to interleukin 33 (IL-33). Whether sST2 can serve as a serum biomarker for HCC is largely unknown.

**Objective.** To investigate the value of sST2 as a serum diagnostic marker for HCC.

**Methods.** This study included 93 newly diagnosed HCC patients (HCC group), 90 chronic hepatitis B patients (CHB group), and 90 healthy individuals (HCs group). Spearman correlation analysis was used to explore the relationships between sST2 and the experimental indicators in HCC group. The receiver operating characteristic (ROC) curve evaluated the efficacy of sST2 alone or in combination with AFP in the diagnosis of HCC.

**Result.** The median level of sST2 was significantly higher in HCC group (24.00 [15.20-49.90] ng/mL) compared to CHB group (19.55 [15.23-24.95] ng/mL) and HCs group (7.65 [5.20-10.53] ng/mL). No significant correlations were found between sST2 and other clinical indicators in HCC group. The Area Under Curve (AUC) of ROC curve to distinguish HCC patients from healthy controls and CHB group was 0.861 (sensitivity 82.80%, specificity 72.10%) and 0.709 (sensitivity 80.60%, specificity 52.50%), respectively. When combined with AFP, the AUC increased to 0.963 (sensitivity 82.90%, specificity 94.20%), and 0.895 (sensitivity 72.0%, specificity 100%), respectively.

**Conclusions.** The serum level of sST2 increased in HCC and its diagnostic performance is comparable to that of AFP, supporting its potential as a promising biomarker for detection of HCC. The combined use of sST2 and AFP enhances diagnostic efficacy for HCC.

## INTRODUCTION

Liver cancer ranks as the sixth most prevalent cancer and the third leading cause of cancer-related mortality globally, with over 900,000 new cases and approximately 830,000 deaths reported in 2020, according to global cancer statistics (*Sung et al., 2021*). Hepatocellular carcinoma (HCC) is the predominant pathological type of liver cancer, comprising about 75% to 85% of cases (*Llovet et al., 2021*). Often, the onset of HCC is insidious, leading many patients to present at advanced stages where treatment options are limited. The 5-year overall survival rate for advanced-stage HCC is approximately 15%, compared to about 75% for early-stage HCC (*Piñero, Dirchwolf & Pessôa, 2020*), underscoring the importance of timely detection for improving patient outcomes. Numerous biomarkers, including embryonic antigens, enzymes and isoenzymes, growth factors and their receptors, cytokines, and protein antigens, are currently being investigated (*Omar et al., 2023*). Although some HCC biomarkers, such as $C-X-C$ motif chemokine ligand 5 (CXCL5), high mobility group box 1 (hmgb1) and vascular endothelial growth factor receptor 1 (VEGFR-1) demonstrate high sensitivity in detection, they show poor correlation with prognosis, and many have not yet been established for routine clinical use (*Zhou et al., 2012*; *Liu et al., 2012*; *Li et al., 2012*; *European Association for the Study of the Liver, 2012*). Thus, finding an ideal biomarker that is universally applicable for routine clinical analysis, and that offers high sensitivity and specificity, as well as ease and speed of detection, is crucial.

Also known as IL1RL1 or IL-1R, ST2 belongs to the interleukin-1 receptor family and exhibits two forms: soluble ST2 (sST2) and transmembrane ST2 (ST2L). The ST2 gene is located on chromosome 2 at 2q11.2 (*Tominaga, Inazawa & Tsuji, 1996*) and is expressed in various organs, including the heart, lungs, kidneys, small intestine, and pancreas (*Mildner et al., 2010*). sST2, initially identified as a cardiac marker, can bind competitively to interleukin 33 (IL-33), thereby inhibiting the IL-33/ST2L signaling pathway and ultimately promoting cardiac remodeling and ventricular dysfunction (*Xing, Liu & Geng, 2021*). *Tang et al. (2016)* found that elevated levels of sST2 were associated with an increased risk of adverse clinical events in acute heart failure and persistently elevated sST2 levels were associated with increased mortality risk. Studies have demonstrated that IL-33 has protective effects against various cardiovascular diseases (CVDs), including cardiac fibrosis, by inducing Th2 cytokines and promoting M2 polarization, whereas sST2 mitigates the biological effects of IL-33 and exacerbates CVDs (*Thanikachalam et al., 2023*). In patients with immune dysregulation disorders such as rheumatoid arthritis (*Kuroiwa et al., 2001*), serum sST2 levels are significantly elevated compared to those in healthy controls. *Oztas et al. (2015)* have suggested the potential of serum sST2 concentration as a predictive biomarker for liver fibrosis in patients infected with hepatitis B virus (HBV). Additionally, IL-33 has been closely associated with hepatitis, liver cirrhosis, and HCC (*Du et al., 2018*; *Sun et al., 2023*; *Wang et al., 2020*). Therefore, it is believed that sST2 may play an important role in HCC

which is closely with inflammation and cirrhosis. However, its diagnostic value in HCC has yet to be fully elucidated. This study aims to explore the potential diagnostic performance of sST2 in the identification of HCC.

## MATERIAL AND METHOD

### Participants

A total of 273 serum samples were collected from three distinct groups: healthy controls (HCs group), patients with chronic hepatitis B (CHB group), and patients with diagnosed hepatocellular carcinoma (HCC group). This study was approved by the Ethics Committee of The First Affiliated Hospital of Sun Yat-sen University ([2020]339) and the ethics committee waived the need for informed consent.

Individuals in the HCs group tested negative for routine blood examinations, biochemical parameters, and HBV-related biomarkers. The diagnosis of chronic hepatitis B (CHB) was confirmed by the presence of hepatitis B surface antigen (HBsAg) for at least 6 months (*Terrault et al., 2018*). Patients with HCC were clinically or pathologically diagnosed according to the American Association for the Study of Liver Diseases guidelines (*Marrero et al., 2018*).

Exclusion criteria included patients with other types of tumors, liver damage due to drugs, autoimmunity, alcohol, parasites, or other microorganisms, other hepatitis virus infections, chronic diseases affecting other body systems, or human immunodeficiency virus (HIV) infection.

### Detection method
#### Complete blood count

A total of 2 ml of peripheral blood samples were collected with EDTA-K2 anticoagulant tubes and then detected for complete blood count (CBC) using a Mindray BC-6800 Plus analyzer (Mindray Diagnostics, Shenzhen, China) with the matching reagents within one hour. The LOT numbers of reagents (Mindray Diagnostics, Shenzhen, China) included 2022032201, 2022032301, 2022031501, 22071803 and 2022070302. CBC composed white blood cell (WBC) count, hemoglobin (Hb), and platelet (PLT) count. The counts of WBC and PLT were detected by Sheath flow impedance combined with flow cytometry, and colorimetric method was used to detect Hb content.

#### Blood biochemical examination

Serum separator tubes were used to collect 4 mL of peripheral blood samples. Samples were centrifuged at 3,500 r/min for 8 min at room temperature, then measured using a Beckman-Coulter AU5800 analyzer (Beckman Coulter, Inc. California, USA) with the matching reagents within two hours. Blood biochemical examination, including enzymatic activities for aspartate aminotransferase (AST) (LOT: AUZ0851, Beckman-Coulter), alanine aminotransferase (ALT) (LOT: AUZ0745, Beckman-Coulter), gamma-glutamyl transferase (GGT) (LOT: AUZ0770, Beckman-Coulter), albumin (ALB) (Lot: AUZ0914, Beckman-Coulter) and total bilirubin (TBIL) (LOT: AUZ0858, Beckman-Coulter) was measured using wet chemistry method.

### Coagulation tests

Coagulation tests were assessed after collecting 2.7 mL of peripheral blood samples with vacuum tubes (containing 1:9 0.109 M trisodium citrate) and centrifuging at 3000 r/min for 10 min at room temperature, using a STAGO STA-R-MAX analyzer (Diagnostica Stago SAS, Asnieres-sur-Seine, France) with the matching reagents within two hours. Paramagnetic particle method was used to measure the coagulation which encompassed prothrombin time (PT) (LOT: 262815, STAGO), international normalized ratio (INR), activated partial thromboplastin time (APTT) (LOT: 262454, STAGO), thrombin time (TT) (LOT: 261703, STAGO), and fibrinogen (FIB) (LOT: 262492, STAGO).

### Hepatitis B virus DNA

Samples collection and centrifugation of hepatitis B virus DNA (HBV-DNA) detection referred to blood biochemical examination. HBV-DNA was quantified with an ABI 7500 Fluorescence quantitative PCR with HBV-DNA kit (LOT: 20220021, DaAn Gene Co, Ltd., of Sun Yat-sen University, China).

### AFP

Samples collection and centrifugation of AFP detection referred to blood biochemical examination. AFP levels were detected using Alinity I analyzers (Abbott Diagnostics, Chicago, USA) with ATP kit (LOT: 39072FN01, Abbott), using the method of chemiluminescent particle immunoassay.

### sST2

Samples collection and centrifugation of sST2 detection referred to blood biochemical examination. Immunofluorescence dry quantitative assay was performed to measure sST2 levels, using a JET-iStar 3000 analyzer with sST2 kit (LOT: ST22203001F, JOINSTAR, Hangzhou, China).

## Statistical analysis

All the detected data represent the mean $\pm$ SD from three independent experiments for each condition. Statistical analyses were conducted using SPSS version 25.0 (IBM Corp., Armonk, NY, USA). A $P$-value of less than 0.05 was considered statistically significant. All data were determined non-normally distributed by the D'Agostino-Pearson omnibus normality test. Kruskal-Wallis test was applied for analyses involving more than two groups. Categorical variables were compared using the Chi-squared test. ROC curves were generated, and AUC values were calculated. Youden's index identified optimal cut-off points. Spearman correlation analyses were used to explore relationships between sST2 and other variables.

## RESULT

### Characteristics of HCC group

The characteristics of the HCC group are presented in Table 1. In HCC group, 87.1% patient were male and over half of the patient (49, 52.7%) were at age of $\geq$60. Based on the widely used and clinically accepted cutoff values (*Galle et al., 2019*), serum levels $\geq 20 \mu g/L$

**Table 1** **Characteristics of HCC group.**

| Feature | | No. of cases (%) |
|---|---|---|
| Gender | Male | 81 (87.1) |
| | Female | 12 (12.9) |
| Age (years) | ≥60 | 49 (52.7) |
| | <60 | 44 (47.3) |
| AFP | − | 39 (41.9) |
| | + | 54 (58.1) |
| TS (cm) | >3 | 69 (74.2) |
| | ≤3 | 24 (25.8) |
| TNM stage | I | 59 (63.4) |
| | II, III, IV | 34 (36.6) |
| PVTT | YES | 24 (25.8) |
| | NO | 69 (74.2) |
| LNM | YES | 17 (18.3) |
| | NO | 76 (81.7) |
| LC | YES | 42 (45.2) |
| | NO | 51 (54.8) |
| HBV infection | YES | 72 (77.4) |
| | NO | 21 (22.6) |

Notes.

TS, tumor size; TNM stage, tumor node metastases stage; PVTT, portal vein tumor thrombus; LNM, lymph node metastasis; LC, liver cirrhosis; HBV, hepatitis B virus.

for AFP was considered positive. At the 20µg/L cutoff, 54 (58.1%) HCC samples were AFP positive. Nearly 75% (69, 74.2%) of the patients were with tumor sizes more than 3 centimeter in diameters. 34 (36.6%) of the patients were stratified into the TNM stage II, III, IV. The portal vein tumor thrombus (PVTT) and lymph node metastases were observed in 24 (25.8%) and 17 (18.3%) patients, respectively. Liver cirrhosis and hepatitis B virus infection were detected in 42 (45.2%) and 72 (77.4%) of patients, respectively.

## HCC group displayed higher sST2

As shown in Fig. 1A, the ages (years) in HCs (56.42 ± 8.74), CHB (58.14 ± 9.19) and HCC (59.23 ± 11.58) group were similar ($P > 0.05$). There was no significant difference in gender (male) percentage among HCs (70, 77.8%), CHB (71, 78.9%) and HCC (81, 87.1%) group ($P > 0.05$) (Fig. 1B). AST showed an elevated level in HCC group (36.00 [28.00–62.00] U/L) compared with HCs group (19.00 [16.00–25.00] U/L, P <0.0001) and CHB group (26.00 [18.00–35.00] U/L, P <0.01) (Fig. 1C). ALT level was higher in HCC group (29.00 [19.00–47.00] U/L) than HCs group (15.5 [13.00–24.00] U/L, P <0.0001) and CHB group (28.00 [21.00–36.00] U/L, P <0.05) (Fig. 1D). LogAFP was also significantly higher in HCC group (71.72 [5.16–839.04] µg/L) than in HCs group (3.94 [2.86−5.40] µg/L, P <0.0001) and CHB group (4.53 [3.12−5.47] µg/L, P <0.0001) (Fig. 1E). The levels of sST2 in HCC group (24.00 [15.20–49.90] ng/mL) increased significantly compared with HCs group (7.65 [5.20–10.53] ng/mL, P <0.0001) and CHB group (19.55 [15.23–24.95] ng/mL, P <0.0001) (Fig. 1F). Besides, AFP levels (Fig. S1a) and sST2 levels (Fig. S1b)
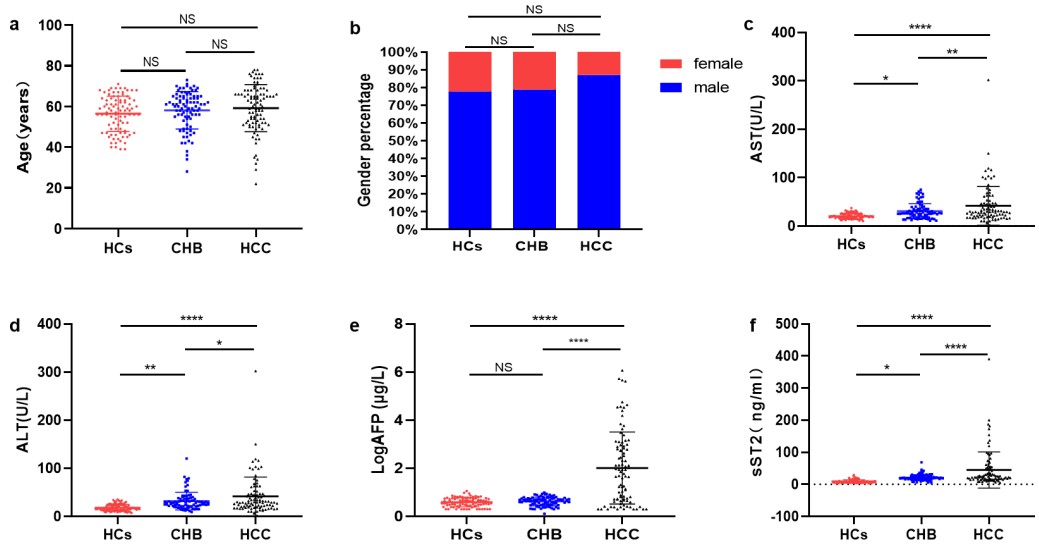

**Figure 1** **HCC group displayed higher sST2.** (A) The comparison of age levels among HCs group, CHB group and HCC group. (B) The comparation of gender percentages among HCs group, CHB group and HCC group. (C) The comparison of AST levels among HCs group, CHB group and HCC group. (D) The comparation of ALT levels among HCs group, CHB group and HCC group. (E) The comparation of LogAFP levels among HCs group, CHB group and HCC group. (F) The comparison of sST2 levels among HCs group, CHB group and HCC group. * ($P < 0.05$), ** ($P < 0.01$), **** ($P < 0.000$).

between the TNM stage I and II–IV in HCC group showed no significant differences ($P > 0.05$), and sST2 levels didn't differ between the patients aged ≥60y and aged <60y (Fig. S1c) or between male and female (Fig. S1d) in HCC group ($P > 0.05$).

## sST2 had no significant correlation with other indicators in HCC group

Further investigation of whether sST2 is associated with other indicators was conducted using Spearman correlation analysis in patients with HCC. As is shown in Table 2, sST2 had no significant correlation with age, AFP, TS, WBC count, Hb, PLT count, ALB, TBIL, ALT, AST, GGT, HBV-DNA copies, PT, APTT, or INR in the whole HCC group ($P > 0.05$). After dividing the HCC group into AFP-negative (AFP-neg) group and AFP-positive (AFP-pos) group, it was found that sST2 was positively correlated with age in the AFP-neg group ($P < 0.01$). There was no significant correlation between sST2 and the aforementioned indicators in AFP-pos group ($P > 0.05$).

## sST2 had a good diagnostic performance which was significantly improved with combination of AFP

To assess the predictive value of sST2 for HCC, the ROC curves were evaluated (Fig. 2 and Table 3). Based on the comparison between the HCC group and the HCs group, the AUC for AFP alone was 0.883 (95% CI [0.830–0.935]) with the optimal cut-off value of 4.575 ng/ml (76.30% sensitivity and 96.50% specificity). The AUC for sST2 alone was 0.861 (95% CI [0.810–0.913]) with the optimal cut-off value of 13.95 μg/L (82.80% sensitivity and 72.10% specificity), indicating similar performance (Fig. 2A). The combination of sST2 and AFP yielded an area under the curve (AUC) of 0.963 (95% CI [0.939–0.987])

**Table 2  The correlation coefficients between sST2 and other indicators in HCC groups.** sST2 had no significant correlation with other indicators in HCC group.

| indicators | sST2 | | |
|---|---|---|---|
| | HCC group (n = 93) | AFP-Pos HCC group (n = 54) | AFP-Neg HCC group (n = 39) |
| **Age** (years) | −0.15 | −0.11 | **0.36**** |
| **AFP** (μg/L) | −0.04 | −0.23 | 0.11 |
| **TS** (cm) | 0.03 | 0.04 | −0.04 |
| **WBC** (*10 9/L) | 0.07 | −0.02 | 0.17 |
| **Hb** (g/L) | 0.03 | −0.12 | 0.25 |
| **PLT** (*10 9/L) | −0.04 | 0.02 | −0.22 |
| **ALB** (g/L) | −0.01 | −0.07 | 0.1 |
| **TBIL** (μmol/L) | 0.09 | 0.01 | 0.19 |
| **ALT** (U/L) | 0.16 | 0.27 | 0 |
| **AST** (U/L) | 0 | 0.04 | −0.08 |
| **GGT** (U/L) | −0.03 | −0.01 | −0.06 |
| **HBV-DNA** (IU/mL) | 0.06 | 0.07 | 0.1 |
| **PT** (s) | 0.02 | 0.04 | 0.02 |
| **INR** | 0.07 | 0.1 | 0.02 |
| **APTT** (s) | 0.03 | −0.06 | 0.16 |
| **TT** (s) | −0.07 | −0.11 | −0.01 |
| **FIB** (g/L) | −0.01 | 0.03 | −0.13 |

with a sensitivity of 89.2% and a specificity of 94.2% for distinguishing HCC group from HCs group. When AFP was negative(AFP $\leq 20\mu$g/L), the AUC for sST2 alone was 0.868 (95% CI [0.795–0.930]), with a threshold concentration of 14.50μg/L (sensitivity 82.10%, specificity 73.30%) (Fig. 2B).

In comparisons between the HCC group and the CHB group, the AUC for AFP alone was 0.8478 (95% CI [0.784–0.911]) with the optimal cut-off value 6.48 ng/ml(sensitivity 69.90% and specificity 97.50%), and for sST2 along the AUC was 0.7094 (95% CI [0.619–0.800]) with the optimal cut-off value 8.45 μg/L(sensitivity 80.6% and specificity 52.5%), as depicted in Fig. 2C. The AUC increased to 0.896 (95% CI [0.845–0.947]) with a sensitivity of 72.00% and a specificity of 100.00% after the combination of AFP and sST2, which distinguished HCC group from CHB group. The AUC for sST2 when AFP was negative was 0.707 (95% CI [0.593–0.821]), with a threshold concentration of 14.45 μg/L (sensitivity 82.10%, specificity 34.60%) (Fig. 2D).

# DISCUSSION

HCC is a significant contributor to cancer-related mortality globally, with its incidence and mortality rates rising over recent decades, posing a serious societal burden. Imaging examinations and the tumor marker AFP are the most widely used diagnostic methods worldwide, yet they have limited accuracy in detecting HCC (*Sauzay et al., 2016*; *Colli et al., 2006*). Only a small percentage (10–20%) of early-stage HCC cases exhibit abnormal

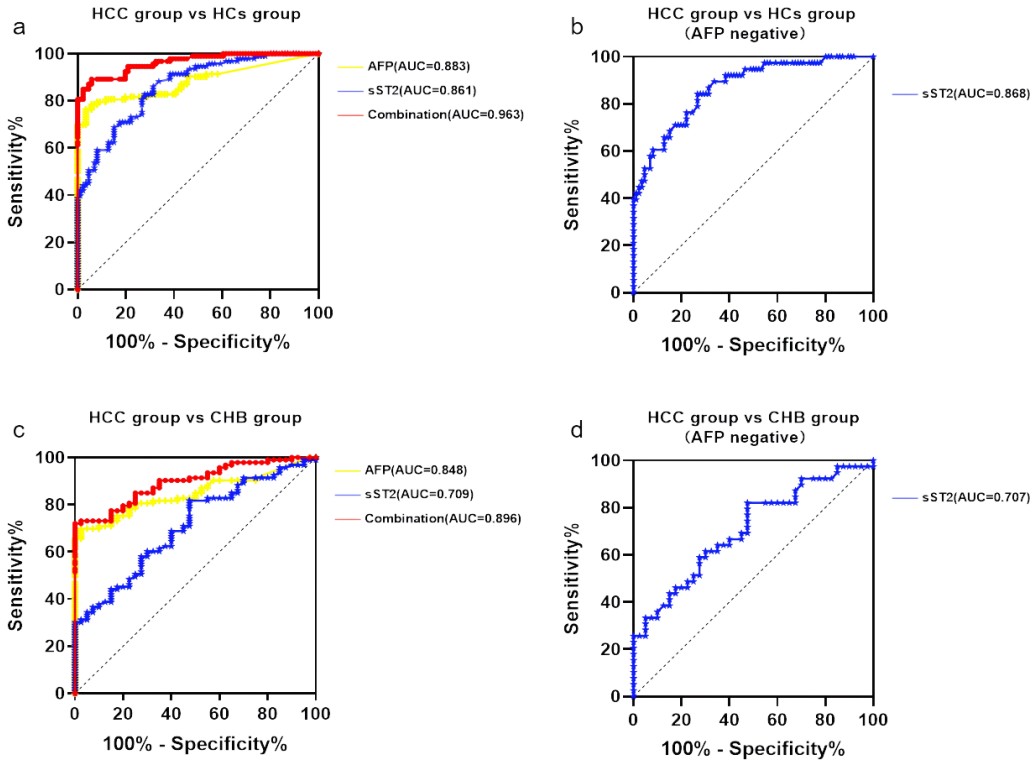

**Figure 2** **SST2 had a good diagnostic performance which significantly elevated with combination of AFP.** (A) ROC curve of sST2, AFP and the combination of sST2 and AFP basing on the comparison between HCC group and HCs group. (B) ROC curve of sST2 basing on the comparison between HCC group and HCs group when AFP was negative. (C) ROC curve of sST2, AFP basing on the comparison between HCC group and CHB group. (D) ROC curve of sST2 basing on the comparison between HCC group and CHB group when AFP was negative.

AFP serum levels, and patients with active liver inflammation may present false-positive AFP results (*European Association for the Study of the Liver, 2018*; *Johnson et al., 2022*). Additionally, while some newly discovered markers have emerged, they still lack reproducibility (*Pinto et al., 2020*). Therefore, it is crucial to seek more reliable biomarkers, which could benefit HCC detection.

Our study focused on sST2 to explore its diagnostic efficacy in HCC. We measured serum sST2 concentrations in healthy controls, patients with chronic hepatitis, and HCC patients, finding significantly higher sST2 levels in the latter group and a moderate increase in those with chronic hepatitis B. When compared with the traditional serum marker AFP in HCC group and HCs group, the diagnostic AUCs of sST2 (AUC = 0.861) was similar to AFP (AUC = 0.883), showing the potential of sST2 as a serum marker of HCC. When AFP was negative, sST2 showed better diagnostic performances in distinguishing HCC from the healthy (AUC = 0.868), but poorer from chronic hepatitis B (AUC = 0.707). The combination of AFP and sST2 significantly enhanced diagnostic performance, with AUCs of 0.963 for distinguishing HCC from healthy controls and 0.896 for distinguishing HCC from chronic hepatitis B. Studies have shown that combinations of biomarkers

**Table 3  The diagnosis of sST2 and AFP.**

| Biomarkers | AUC (95%CI) | Sensitivity(%) | Specificity(%) |
|---|---|---|---|
| **HCC vs HCs** | | | |
| AFP | 0.883 (0.830–0.935) | 76.30 | 96.50 |
| sST2 | 0.861 (0.810–0.913) | 82.80 | 72.10 |
| Combination | 0.963 (0.939–0.987) | 82.90 | 94.20 |
| **HCC (AFP negative) vs HCs** | | | |
| sST2 | 0.868 (0.795–0.930) | 82.10 | 73.30 |
| **HCC vs CHB** | | | |
| AFP | 0.848 (0.784–0.911) | 69.90 | 97.50 |
| sST2 | 0.709 (0.619–0.800) | 80.60 | 52.50 |
| Combination | 0.896 (0.845–0.947) | 72.00 | 100.00 |
| **HCC (AFP negative) vs CHB** | | | |
| sST2 | 0.707 (0.593–0.821) | 82.10 | 34.60 |

provide a higher diagnostic value than single biomarkers (*Williams, 2009*). The AUC of AFP in diagnosing HCC ranges between 0.80 and 0.85; this can increase when AFP is combined with other markers, such as 0.9675 (AFP + chemokine 20) (*Deng et al., 2024*), 0.876 (AFP + osteopontin) (*Zhu et al., 2020*), 0.94 (AFP + aberrant sialylated N-glycans) (*Zhu et al., 2024*) and 0.96 (AFP + CXC chemokine receptor 2, C-C motif chemokine receptor-2 and chromatin regulators EP400)(*Shi et al., 2014*). Compared with these values, our study suggests that the combination of sST2 and AFP is comparable or even superior, indicating the potential of this combination for HCC screening. Besides, sST2 may serve as an independent detection index of HCC for having no correlation with other indicators.

This study has several limitations. Firstly, this is a single-center study with limited number of cases analyzed and the finding should be further confirmed by a prospective multicenter study with a larger number of patients. Second, the relationship between sST2 levels and patient prognosis requires further exploration.

## CONCLUSION

In summary, our study highlights the potential of sST2 as a blood-based marker for HCC detection and the enhanced diagnostic performance of combined sST2 and AFP use in HCC screening, suggesting that further research could lead to novel strategies for managing HCC.

### Funding

This work was supported by grants from National Natural Science Foundation of China (Grant Nos. 81972750, 82373069), the Guangdong Basic and Applied Basic Research Foundation (Grant Nos. 2024A1515011089, 2022A1515220130) and Guangzhou Science and Technology Plan Project (Grant No 2024A04J6489). The funders had no role in study design, data collection and analysis, decision to publish, or preparation of the manuscript.

## Grant Disclosures

The following grant information was disclosed by the authors:

National Natural Science Foundation of China: 81972750, 82373069.

The Guangdong Basic and Applied Basic Research Foundation: 2024A1515011089, 2022A1515220130.

Guangzhou Science and Technology Plan Project: 2024A04J6489.

## Competing Interests

The authors declare there are no competing interests.

## Author Contributions

- Xiuxin Tang performed the experiments, analyzed the data, prepared figures and/or tables, and approved the final draft.
- Dong Wang analyzed the data, prepared figures and/or tables, and approved the final draft.
- Tangdan Ding analyzed the data, prepared figures and/or tables, and approved the final draft.
- Rongqi Lin performed the experiments, prepared figures and/or tables, and approved the final draft.
- Meifang He performed the experiments, prepared figures and/or tables, and approved the final draft.
- Ruizhi Wang conceived and designed the experiments, authored or reviewed drafts of the article, and approved the final draft.
- Liubing Li conceived and designed the experiments, prepared figures and/or tables, authored or reviewed drafts of the article, and approved the final draft.

## Human Ethics

The following information was supplied relating to ethical approvals (*i.e.*, approving body and any reference numbers):

This study was approved by the Ethics Committee of The First Affiliated Hospital of Sun Yat-sen University ([2020] 339).

## Data Availability

The raw measurements are available in the Supplementary File.

## Supplemental Information

Supplemental information for this article can be found online at http://dx.doi.org/10.7717/peerj.18142#supplemental-information.

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
