# Peer review of "Assessment of combined serum sST2 and AFP levels in the diagnosis of hepatocellular carcinoma"

_PeerJ, doi:10.7717/peerj.18142_

## Round 0.1 · original submission · Major Revisions

In this manuscript, the authors studied sST2 as a potential biomarker for hepatocellular carcinoma (HCC), thus potentially benefiting future studies and HCC patients. Yet, some additional clarification/discussion would be really appreciated. Kindly see the details below and comments from our reviewers.

As indicated by the authors in lines 51-53, some HCC biomarkers have “demonstrated high sensitivity in detection,” yet, “they show poor correlation with prognosis.” What are these biomarkers? Do these biomarkers include AFP? Accordingly, it would also be of interest to see if the sST2 level with or without combination with AFP shows a better prognosis correlation. It is understandable that the corresponding clinical treatment outcomes may not be available. In this situation, it would be interesting to see if the sST2 level with or without combination with AFP shows a better correlation with the patients’ TNM stages, at least between I and II-IV.

It seems that some parts of the Discussion section overlap with the Introduction, such as the ST2 research by other groups. Accordingly, it is suggested that the background information be kept in the Introduction and that the obtained results be compared with previous findings in the Discussion.

It is further suggested to elaborate on how the authors arrived at the 20 µg/L as the AFP cutoff in line 108.

Additionally, kindly expand the Material and Method section as indicated by some of our reviewers.

Some very minor formality/grammatical issues were noticed while reviewing:
(1) There is a duplicate “.” in line 84.
(2) There is a space missing in the term “Coagulation indiceswere” in line 90.
(3) To maintain consistency with the subsections, it is suggested that the sections be numbered, too.

(4) In the figure legend for Figure 1, it seems the authors meant to say “elevated” instead of “evaluated.”

(5) Kindly amend “Hcs” in the figures to “HCs” for consistency with the text.

(6) If appropriate, it is suggested to change the “associate” in line 128 to “associated.”

(7) An “or” is missing before “INR” in line 131.

(8) It seems Table 4 is not discussed in the text. Is the “Table 3” in line 139 meant to be Table 4?

(9) It is suggested to provide the full names of the acronyms upon their first appearance and continue to use the acronyms from there, such as “AFP” in lines 19 and 160, “ST2” in lines 2, 56 and 184, as well as “HCC” in lines 18 and 158.

·

Basic reporting

1. Overall, the language of this manuscript is clear. Although, some refinement is needed to make it more professional and scientific (e.g. line 20, line 51-53, line 115-116).

2. Reference is provided clearly and sufficiently, except some references are needed for line 42-44.

3. The overall structure of the article meets the requirements of a scientific publication. Raw data can be found in the supplement table. The presentation of the data may require some changes for better visualization and story-telling.

i. Some of the data that were presented as table can be presented as figure for better visualization, including but not limited to the level of AST, CHB, ST2. The visualization should show the statistical parameters, such as mean/median, standard error, and p-value, as well as each data point.

4. Results addressed the questions raised by the hypothesis.

Experimental design

1. In detection method part, more details are needed to ensure reproducibility. For example, all the reagents information, primer sequence for PCR should be listed. Also, the procedure of how experiments were performed needs to be documented in detail. I suggest the author write multiple sections for each method applied in this study under detection method.
2. In Statistical analysis, more details are needed. For example, for the t-test, whether the author use paired or unpaired needs to be documented in the manuscript.
3. In method, the author stated “Both Spearman and Pearson correlation analyses were used to explore relationships between sST2 and other variables”, while in the result section, only Spearman correlation analysis is mentioned.

Validity of the findings

This manuscript meets the requirements with some improvements mentioned in the previous sections.

Reviewer 2 ·

Basic reporting

1. The author should consider using color blind friendly palette for Figure 1 to accommodate need from diverse readers (i.e. avoid red and green combination).
2. The author should plot the individual data points of results in Table 2 with data average, error bar and p value on the plot for better visualization purpose.
3. The author should improve the quality of Table 3, and format table 3 better for reader to read each value clearly.

Experimental design

Sample size of each group should be similar to enable a fair comparison and generate valid statistical analysis. The author mentioned that "This study included 93 newly diagnosed HCC patients (HCC group), 40 chronic hepatitis B patients (CHB group), and 86 healthy individuals (HCs group)" in Methods section (line 25-26). However, sample size of each group should be similar to enable a fair comparison and generate valid statistical analysis. The significance observed in Result could be due to the sample size discrepancy. Also, the characteristics (for example, age, gender percentage etc.) should be similar across different group to enable a fair comparison.

Validity of the findings

The author should provide more detailed information about the statistics and reproducibility of the data in statistical analysis section (Line 96 ). Specifically, the author should claim that all data represent the mean ± SEM or mean ± SD from ≥3 independent experiments (independent biological replicas) for each condition. The D’Agostino-Pearson omnibus normality test should be used to determine whether data are normally distributed. Datasets with gaussian distributions should be compared using Student’s t-test (two-tailed) or one-way ANOVA followed by Tukey’s post hoc test. For comparing non-Gaussian distributions, the nonparametric Mann–Whitney U test or Kruskal–Wallis (with post hoc Dunn) should be used for comparisons between two or more groups, respectively. The exact p-values and how stars were defined should be included in all figure or table legends.

Additional comments

In this manuscript, Tang et al. investigated the potential diagnostic performance of sST2 in the identification of HCC. The major finding of this manuscript is that the serum level of sST2 increased in HCC patient and its diagnostic performance is comparable to that of AFP, indicating its potential as a promising biomarker for detection of HCC. Moreover the combined use of sST2 and AFP enhances diagnostic efficacy for HCC, providing insights into the diagnose of HCC. However, additional data and analysis are required to support and validate their findings.

Major Concerns:
1. The descriptive nature and absence of mechanistic information represent the major weaknesses of this manuscript. Although the reviewer appreciates the observations showing that high sST2 expression was associated with HCC diagnose in patients, the authors did not delve into the molecular biology and signaling pathway of the process.

Reviewer 3 ·

Basic reporting

no comment

Experimental design

See the first bullet point in the additional comment.

Validity of the findings

no comment

Additional comments

The primary objective of the manuscript is to determine whether soluble ST2 (sST2) can serve as a biomarker for diagnosing Hepatocellular Carcinoma (HCC). To explore this, sST2 levels were measured in serum samples from three groups: healthy volunteers, chronic hepatitis B patients, and HCC patients. Statistical analyses were then performed on the collected data. Overall, the manuscript is well-written, with clear articulation of both the objective and the experimental results. Here are a few minor comments:
1. The detection method lacks sufficient detail for replication. While the procedures for detecting white blood cells (WBC), hemoglobin (Hb), platelet (PLT) counts, sST2 and other parameters are well-established in the hospital, it would be beneficial for the authors to specify the principles, reagents, and step-by-step procedures used. This additional information would aid others in replicating the study.
2. In Table 2, the authors compare sST2 levels among healthy controls (HCs), chronic hepatitis B patients (CHB), and Hepatocellular Carcinoma patients (HCC). I am curious if the authors have analyzed sST2 levels by gender and age. The table suggests that the HCC group consists of a higher proportion of older patients and males. Could the higher overall sST2 levels in HCC group be due to higher levels of sST2 in older individuals and males?
3. In Table 3, the number of patients in the AFP-Positive HCC group is 54, and in the AFP-Negative HCC group, it is 39. However, in Table 1, there are a total of 22 AFP-negative patients and 71 AFP-positive patients. Could the authors clarify these discrepancies?

---

## Round 0.2 · Minor Revisions

Many thanks for the authors’ amendments and responses. Much appreciated.
There are only some very minor formality/consistency issues, as pointed out by our reviewers and noted below. Kindly make appropriate changes accordingly, and we should be ready to publish the manuscript here.

(1) There is a duplicated “,” in line 22.
(2) If appropriate, kindly change “detected CBC tests” in line 100 to “detected for CBC” or similar.
(3) It seems “detedct” in line 105 is meant to be “detected” while “dectece” in line 106 is “detect.”
(4) The “tests” in line 120 and some other parts of Section 2.2 seems unnecessary.
(5) It seems a “was” is missing before “significantly” in line 180 and in the title of Table 3.
(6) Kindly amend “a independent” in line 229 to “an independent.”
(7) Kindly change “Hcs” in Table 3 to “HCs” for consistency.
(8) Kindly change “Lo'gAFP” in the figure legend for Figure 1 to “LogAFP.” In the same figure legend, kindly change “level” to “levels” and “gender percentage” to “gender percentages” if appropriate.

·

Basic reporting

The revised manuscript has been successfully addressed the issues. The manuscript is in good shape.

Minor revision is needed:
1. Figure 1b. The Y-axis labelling should change to a 0-100 scale since it is the percentage.

Experimental design

No comment

Validity of the findings

No comment

Reviewer 2 ·

Basic reporting

No further comments upon review

Experimental design

No further comments upon review

Validity of the findings

No further comments upon review

Additional comments

No further comments upon review

Reviewer 3 ·

Basic reporting

no comment

Experimental design

no comment

Validity of the findings

no comment

Additional comments

Revision has been received, and all my questions have been addressed. However, I noticed two minor errors in the manuscript:
• In line 107, there is a typo: “usedt” should be corrected.
• In line 191, the sensitivity is listed as 73.3% and specificity as 82.1%, but these numbers are reversed in Table 3.

---

## Round 0.3 · Minor Revisions

Many thanks for the revisions. Following one of our reviewer's previous comments, kindly update the "0.82" and "0.73" in Table 3 to percentages if appropriate and accurate. Then it should be ready for publishing.

---

## Round 0.4 · accepted · Accept

Many thanks for your rapid reply.